# Latent Attention For If-Then Program Synthesis

**Xinyun Chen**[*]
Shanghai Jiao Tong University

**Chang Liu**　**Richard Shin**　**Dawn Song**
UC Berkeley

**Mingcheng Chen**[†]
UIUC

## Abstract

Automatic translation from natural language descriptions into programs is a long-standing challenging problem. In this work, we consider a simple yet important sub-problem: translation from textual descriptions to If-Then programs. We devise a novel neural network architecture for this task which we train end-to-end. Specifically, we introduce Latent Attention, which computes multiplicative weights for the words in the description in a two-stage process with the goal of better leveraging the natural language structures that indicate the relevant parts for predicting program elements. Our architecture reduces the error rate by $28.57\%$ compared to prior art [3]. We also propose a one-shot learning scenario of If-Then program synthesis and simulate it with our existing dataset. We demonstrate a variation on the training procedure for this scenario that outperforms the original procedure, significantly closing the gap to the model trained with all data.

## 1 Introduction

A touchstone problem for computational linguistics is to translate natural language descriptions into executable programs. Over the past decade, there has been an increasing number of attempts to address this problem from both the natural language processing community and the programming language community. In this paper, we focus on a simple but important subset of programs containing only one If-Then statement.

An If-Then program, which is also called a *recipe*, specifies a *trigger* and an *action* function, representing a program which will take the action when the trigger condition is met. On websites, such as IFTTT.com, a user often provides a natural language description of the recipe's functionality as well. Recent work [16, 3, 7] studied the problem of automatically synthesizing If-Then programs from their descriptions. In particular, LSTM-based sequence-to-sequence approaches [7] and an approach of ensembling a neural network and logistic regression [3] were proposed to deal with this problem. In [3], however, the authors claim that the diversity of vocabulary and sentence structures makes it difficult for an RNN to learn useful representations, and their ensemble approach indeed shows better performance than the LSTM-based approach [7] on the function prediction task (see Section 2).

In this paper, we introduce a new attention architecture, called *Latent Attention*, to overcome this difficulty. With Latent Attention, a weight is learned on each token to determine its importance for prediction of the trigger or the action. Unlike standard attention methods, Latent Attention computes the token weights in a two-step process, which aims to better capture the sentence structure. We show that by employing Latent Attention over outputs of a bi-directional LSTM, our new Latent Attention model can improve over the best prior result [3] by 5 percentage points from $82.5\%$ to $87.5\%$ when predicting the trigger and action functions together, reducing the error rate of [3] by $28.57\%$.

Besides the If-Then program synthesis task proposed by [16], we are also interested in a new scenario. When a new trigger or action is released, the training data will contain few corresponding

---

[*]Part of the work was done while visiting UC Berkeley.
[†]Work was done while visiting UC Berkeley. Mingcheng Chen is currently working at Google [X].

examples. We refer to this case as a *one-shot learning* problem. We show that our Latent Attention model on top of dictionary embedding combining with a new training algorithm can achieve a reasonably good performance for the one-shot learning task.

## 2   If-Then Program Synthesis

**If-Then Recipes.**   In this work, we consider an important class of simple programs called *If-Then"recipes"* (or recipes for short), which are very small programs for event-driven automation of tasks. Specifically, a recipe consists of a trigger and an action, indicating that the action will be executed when the trigger is fulfilled.

The simplicity of If-Then recipes makes it a great tool for users who may not know how to code. Even non-technical users can specify their goals using recipes, instead of writing code in a more full-fledged programming language. A number of websites have embraced the If-Then programming paradigm and have been hugely successful with tens of thousands of personal recipes created, including IFTTT.com and Zapier.com. In this paper, we focus on data crawled from IFTTT.com.

IFTTT.com allows users to share their recipes publicly, along with short natural language descriptions to explain the recipes' functionality. A recipe on IFTTT.com consists of a *trigger channel*, a *trigger function*, an *action channel*, an *action function*, and arguments for the functions. There are a wide range of channels, which can represent entities such as devices, web applications, and IFTTT-provided services. Each channel has a set of functions representing events (i.e., trigger functions) or action executions (i.e., action functions).

For example, an IFTTT recipe with the following description

<div align="center">Autosave your Instagram photos to Dropbox</div>

has the trigger channel `Instagram`, trigger function `Any_new_photo_by_you`, action channel `Dropbox`, and action function `Add_file_from_URL`. Some functions may take arguments. For example, the `Add_file_from_URL` function takes three arguments: the source URL, the name for the saved file, and the path to the destination folder.

**Problem Setup.**   Our task is similar to that in [16]. In particular, for each description, we focus on predicting the channel and function for trigger and action respectively. Synthesizing a valid recipe also requires generating the arguments. As argued by [3], however, the arguments are not crucial for representing an If-Then program. Therefore, we defer our treatment for arguments generation to the supplementary material, where we show that a simple frequency-based method can outperform all existing approaches. In this way, our task turns into two classification problems for predicting the trigger and action functions (or channels).

Besides the problem setup in [16], we also introduce a new variation of the problem, a one-shot learning scenario: when some new channels or functions are initially available, there are very few recipes using these channels and functions in the training set. We explore techniques to still achieve a reasonable prediction accuracy on labels with very few training examples.

## 3   Related Work

Recently there has been increasing interests in executable code generation. Existing works have studied generating domain-specific code, such as regular expressions [12], code for parsing input documents [14], database queries [22, 4], commands to robots [10], operating systems [5], smartphone automation [13], and spreadsheets [8]. A recent effort considers translating a mixed natural language and structured specification into programming code [15]. Most of these approaches rely on semantic parsing [19, 9, 1, 16]. In particular, [16] introduces the problem of translating IFTTT descriptions into executable code, and provides a semantic parsing-based approach. Two recent work studied approaches using sequence-to-sequence model [7] and an ensemble of a neural network and a logistic regression model [3] to deal with this problem, and showed better performance than [16]. We show that our Latent Attention method outperforms all prior approaches. Recurrent neural networks [21, 6] along with attention [2] have demonstrated impressive results on tasks such as machine translation [2], generating image captions [20], syntactic parsing [18] and question answering [17].

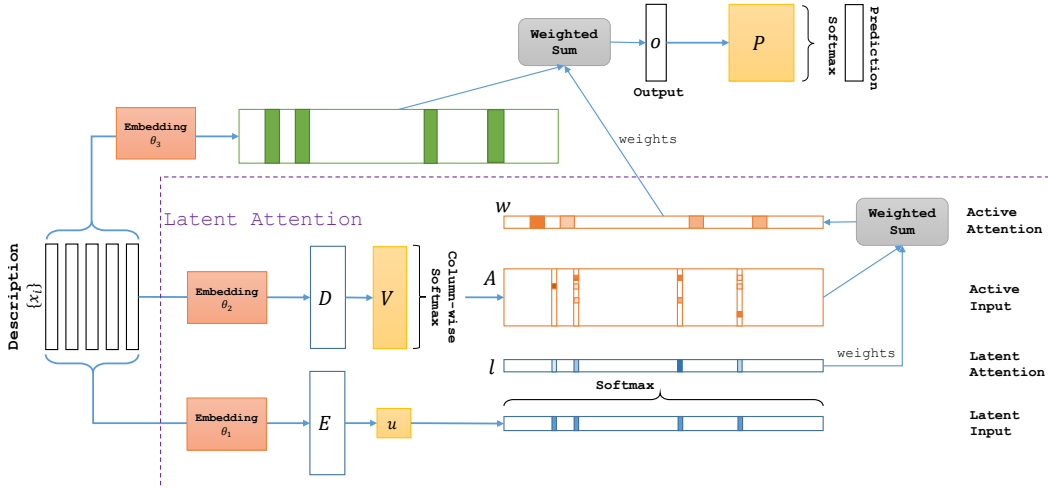

Figure 1: Network Architecture

## 4 Latent Attention Model

### 4.1 Motivation

To translate a natural language description into a program, we would like to locate the words in the description that are the most relevant for predicting desired labels (trigger/action channels/functions). For example, in the following description

Autosave Instagram photos to your Dropbox folder

the blue text "Instagram photos" is the most relevent for predicting the trigger. To capture this information, we can adapt the attention mechanism [2, 17] —first compute a weight of the importance of each token in the sentence, and then output a weighted sum of the embeddings of these tokens.

However, our intuition suggests that the weight for each token depends not only on the token itself, but also the overall sentence structure. For example, in

Post photos in your Dropbox folder to Instagram

"Dropbox" determines the trigger, even though in the previous example, which contains almost the same set of tokens, "Instagram" should play this role. In this example, the prepositions such as "to" hint that the trigger channel is specified in the middle of the description rather than at the end. Taking this into account allows us to select "Dropbox" over "Instagram".

Latent Attention is designed to exploit such clues. We use the usual attention mechanism for computing a *latent weight* for each token to determine which tokens in the sequence are more relevant to the trigger or the action. These latent weights determine the final attention weights, which we call *active weights*. As an example, given the presence of the token "to", we might look at the tokens before "to" to determine the trigger.

### 4.2 The network

The Latent Attention architecture is presented in Figure 1. We follow the convention of using lower-case letters to indicate column vectors, and capital letters for matrices. Our model takes as input a sequence of symbols $x_1, ..., x_J$, with each coming from a dictionary of $N$ words. We denote $X = [x_1, ..., x_J]$. Here, $J$ is the maximal length of a description. We illustrate each layer of the network below.

**Latent attention layer.** We assume each symbol $x_i$ is encoded as a one-hot vector of $N$ dimensions. We can embed the input sequence $X$ into a $d$-dimensional embedding sequence using $E = \texttt{Embed}_{\theta_1}(X)$, where $\theta_1$ is a set of parameters. We will discuss different embedding methods in Section 4.3. Here $E$ is of size $d \times J$.

The latent attention layer's output is computed as a standard softmax on top of $E$. Specifically, assume that $l$ is the $J$-dimensional output vector, $u$ is a $d$-dimensional trainable vector, we have

$$l = \mathbf{softmax}(u^T \mathtt{Embed}_{\theta_1}(X))$$

**Active attention layer.** The active attention layer computes each token's weight based on its importance for the final prediction. We call these weights *active weights*. We first embed $X$ into $D$ using another set of parameters $\theta_2$, i.e., $D = \mathtt{Embed}_{\theta_2}(X)$ is of size $d \times J$. Next, for each token $D_i$, we compute its active attention input $A_i$ through a softmax:

$$A_i = \mathbf{softmax}(V D_i)$$

Here, $A_i$ and $D_i$ denote the the $i$-th column vector of $A$ and $D$ respectively, and $V$ is a trainable parameter matrix of size $J \times d$. Notice that $V D_i = (V D)_i$, we can compute $A$ by performing *column-wise softmax* over $V D$. Here, $A$ is of size $J \times J$.

The active weights are computed as the sum of $A_i$, weighted by the output of latent attention weight:

$$w = \sum_{i=1}^{J} l_i A_i = Al$$

**Output representation.** We use a third set of parameters $\theta_3$ to embed $X$ into a $d \times J$ embedding matrix, and the final output $o$, a $d$-dimensional vector, is the sum of the embedding weighted by the active weights:

$$o = \mathtt{Embed}_{\theta_3}(X)w$$

**Prediction.** We use a softmax to make the final prediction: $\hat{f} = \mathbf{softmax}(Po)$, where $P$ is a $d \times M$ parameter matrix, and $M$ is the number of classes.

### 4.3 Details

**Embeddings.** We consider two embedding methods for representing words in the vector space. The first is a straightforward word embedding, i.e., $\mathtt{Embed}_{\theta}(X) = \theta X$, where $\theta$ is a $d \times N$ matrix and the rows of $X$ are one-hot vectors over the vocabulary of size $N$. We refer to this as "dictionary embedding" later in the paper. $\theta$ is not pretrained with a different dataset or objective, but initialized randomly and learned at the same time as all other parameters. We observe that when using Latent Attention, this simple method is effective enough to outperform some recent results [16, 7].

The other approach is to take the word embeddings, run them through a bi-directional LSTM (BDLSTM) [21], and then use the concatenation of two LSTMs' outputs at each time step as the embedding. This can take into account the context around a token, and thus the embeddings should contain more information from the sequence than from a single token. We refer to such an approach as "BDLSTM embedding". The details are deferred to the supplementary material. In our experiments, we observe that with the help of this embedding method, Latent Attention can outperform the prior state-of-the-art.

In Latent Attention, we have three sets of embedding parameters, i.e., $\theta_1, \theta_2, \theta_3$. In practice, we find that we can equalize the three without loss of performance. Later, we will show that keeping them separate is helpful for our one-shot learning setting.

**Normalizing active weights.** We find that normalizing the active weights $a$ before computing the output is helpful to improve the performance. Specifically, we compute the output as

$$o = \mathtt{Embed}_{\theta}(X)\mathbf{normalized}(w) = \mathtt{Embed}_{\theta}(X)\frac{w}{||w||}$$

where $||w||$ is the $L_2$-norm of $w$. In our experiments, we observe that this normalization can improve the performance by 1 to 2 points.

**Padding and clipping.** Latent Attention requires a fixed-length input sequence. To handle inputs of variable lengths, we perform padding and clipping. If an input's length is smaller than $J$, then we pad it with null tokens at the end of the sequence. If an input's length is greater than $J$ (which is 25 in our experiements), we keep the first 12 and the last 13 tokens, and get rid of all the rest.

**Vocabulary.** We tokenize each sentence by splitting on whitespace and punctuation (e.g., ., !?”′ : ; )( ), and convert all characters into lowercase. We keep all punctuation symbols as tokens too. We map each of the top 4,000 most frequent tokens into themselves, and all the rest into a special token ⟨UNK⟩. Therefore our vocabulary size is 4,001. Our implementation has no special handling for typos.

## 5   If-Then Program Synthesis Task Evaluation

In this section, we evaluate our approaches with several baselines and previous work [16, 3, 7]. We use the same crawler from Quirk et al. [16] to crawl recipes from IFTTT.com. Unfortunately, many recipes are no longer available. We crawled all remaining recipes, ultimately obtaining 68,083 recipes for the training set. [16] also provides a list of 5,171 recipes for validation, and 4,294 recipes for test. All test recipes come with labels from Amazon Mechanical Turk workers. We found that only 4,220 validation recipes and 3,868 test recipes remain available. [16] defines a subset of test recipes, where each recipe has at least 3 workers agreeing on its labels from IFTTT.com, as the gold testset. We find that 584 out of the 758 gold test recipes used in [16] remain available. We refer to these recipes as the *gold test set*. We present the data statistics in the supplementary material.

**Evaluated methods.** We evaluate two embedding methods as well as the effectiveness of different attention mechanisms. In particular, we compare no attention, standard attention, and Latent Attention. Therefore, we evaluate six architectures in total. When using dictionary embedding with no attention, for each sentence, we sum the embedding of each word, then pass it through a softmax layer for prediction. For convenience, we refer to such a process as *standard softmax*. For BDL-STM with no attention, we concatenate final states of forward and backward LSTMs, then pass the concatenation through a softmax layer for prediction. The two embedding methods with standard attention mechanism [17] are described in the supplementary material. The Latent Attention models have been presented in Section 4.

**Training details.** For architectures with no attention, they were trained using a learning rate of 0.01 initially, which is multiplied by 0.9 every 1,000 time steps. Gradients with $L_2$ norm greater than 5 were scaled down to have norm 5. For architectures with either standard attention mechanism or Latent Attention, they were trained using a learning rate of 0.001 without decay, and gradients with $L_2$ norm greater than 40 were scaled down to have norm 40. All models were trained using Adam [11]. All weights were initialized uniformly randomly in $[-0.1, 0.1]$. Mini-batches were randomly shuffled during training. The mini-batch size is 32 and the embedding vector size $d$ is 50.

**Results.** Figure 2 and Figure 3 present the results of prediction accuracy on channel and function respectively. Three previous works' results are presented as well. In particular, [16] is the first work introducing the If-Then program synthesis task. [7] investigates the approaches using sequence-to-sequence models, while [3] proposes an approach to ensemble a feed-forward neural network and a logistic regression model. The numerical values for all data points can be found in the supplementary material.

For our six architectures, we use 10 different random initializations to train 10 different models. To ensemble $k$ models, we choose the best $k$ models on the validation set among the 10 models, and average their softmax outputs as the ensembled output. For the three existing approaches [16, 7, 3], we choose the best results from these papers.

We train the model to optimize for function prediction accuracy. The channel accuracy in Figure 2 is computed in the following way: to predict the channel, we first predict the function (from a list of all functions in all channels), and the channel that the function belongs to is returned as the predicted channel. We observe that

- Latent Attention steadily improves over standard attention architectures and no attention ones using either embedding method.
- In our six evaluated architectures, ensembling improves upon using only one model significantly.
- When ensembling more than one model, BDLSTM embeddings perform better than dictionary embeddings. We attribute this to that for each token, BDLSTM can encode the

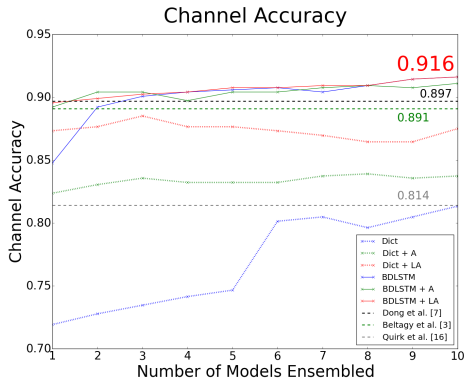

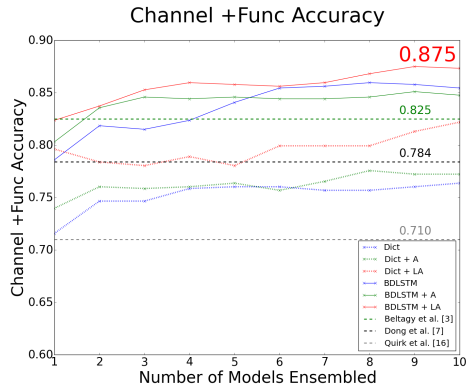

Figure 2: Accuracy for Channel    Figure 3: Accuracy for Channel+Function

information of its surrounding tokens, e.g., phrases, into its embedding, which is thus more effective.

- For the channel prediction task in Figure 2, all architectures except dictionary embedding with no attention (i.e., Dict) can outperform [16]. Ensembling only 2 BDLSTM models with either standard attention or Latent Attention is enough to achieve better performance than prior art [7]. By ensembling 10 BDLSTM+LA models, we can improve the latest results [7] and [3] by 1.9 points and 2.5 point respectively.

- For the function prediction task in Figure 3, all our six models (including Dict) outperform [16]. Further, ensembling 9 BDLSTM+LA can improve the previous best results [3] by 5 points. In other words, our approach reduces the error rate of [3] by 28.57%.

## 6    One-Shot Learning

We consider the scenario when websites such as IFTTT.com release new channels and functions. In such a scenario, for a period of time, there will be very few recipes using the newly available channels and fucntions; however, we would still like to enable synthesizing If-Then programs using these new functions. The rarity of such recipes in the training set creates a challenge similar to the *one-shot learning* setting. In this scenario, we want to leverage the large amount of recipes for existing functions, and the goal is to achieve a good prediction accuracy for the new functions without significantly compromising the overall accuracy.

### 6.1    Datasets to simulate one-shot learning

To simulate this scenario with our existing dataset, we build two one-shot variants of it as follows. We first split the set of trigger functions into two sets, based on their frequency. The top100 set contains the top 100 most frequently used trigger functions, while the non-top100 set contains the rest.

Given a set of trigger functions $S$, we can build a skewed training set to include all recipes using functions in $S$, and 10 randomly chosen recipes for each function not in $S$. We denote this skewed training set created based on $S$ as $(S, \overline{S})$, and refer to functions in $S$ as *majority functions* and functions in $\overline{S}$ as *minority functions*. In our experiments, we construct two new training sets by choosing $S$ to be the top100 set and non-top100 set respectively. We refer to these two training sets as SkewTop100 and SkewNonTop100.

The motivation for creating these datasets is to mimic two different scenarios. On one hand, Skew-Top100 simulates the case that at the startup phase of a service, popular recipes are first published, while less frequently used recipes are introduced later. On the other hand, SkewNonTop100 captures the opposite situation. The statistics for these two training sets are presented in the supplementary material. While SkewTop100 is more common in real life, the SkewNonTop100 training set is only 15.73% of the entire training set, and thus is more challenging.

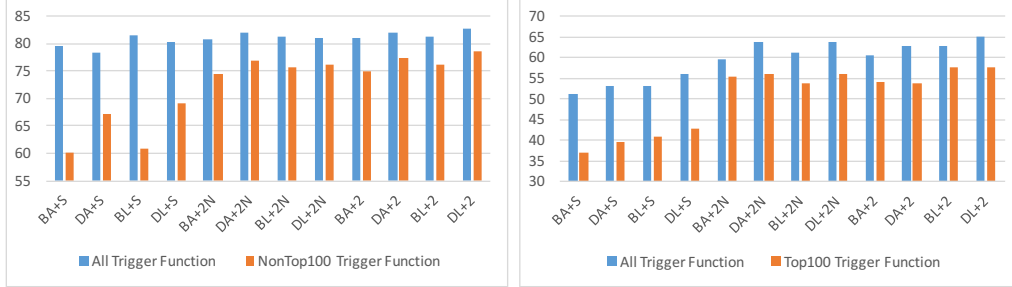

(a) Trigger Function Accuracy (SkewTop100)      (b) Trigger Function Accuracy (SkewNonTop100)

Figure 4: One-shot learning experiments. For each column XY-Z, X from {B, D} represents whether the embedding is BDLSTM or Dictionary; Y is either empty, or is from {A, L}, meaning that either no attention is used, or standard attention or Latent Attention is used; and Z is from {S, 2N, 2}, denoting standard training, naïve two-step training or two-step training.

## 6.2 Training

We evaluate three training methods as follows, where the last one is specifically designed for attention mechanisms. In all methods, the training data is either SkewTop100 or SkewNonTop100.

**Standard training.** We do not modify the training process.

**Naïve two-step training.** We do standard training first. Since the data is heavily skewed, the model may behave poorly on the minority functions. From a training set $(S, \overline{S})$, we create a rebalanced dataset, by randomly choosing 10 recipes for each function in $S$ and all recipes using functions in $\overline{S}$. Therefore, the numbers of recipes using each function are similar in this rebalanced dataset. We recommence the training using this rebalanced training dataset in the second step.

**Two-step training.** We still do standard training first, and then create the rebalanced dataset in the similar way as that in naïve two-step training. However, in the second step, instead of training the entire network, we keep the attention parameters fixed, and train only the parameters in the remaining part of the model. Take the Latent Attention model depicted in Figure 1 as an example. In the second step, we keep parameters $\theta_1$, $\theta_2$, $u$, and $V$ fixed, and only update $\theta_3$ and $P$ while training on the rebalanced dataset. We based this procedure on the intuition that since the rebalanced dataset is very small, fewer trainable parameters enable easier training.

## 6.3 Results

We compare the three training strategies using our proposed models. We omit the no attention models, which do not perform better than attention models and cannot be trained using two-step training. We only train one model per strategy, so the results are without ensembling. The results are presented in Figure 4. The concrete values can be found in the supplementary material. For reference, the best single BDLSTM+LA model can achieve $89.38\%$ trigger function accuracy: $91.11\%$ on top100 functions, and $85.12\%$ on non-top100 functions. We observe that

- Using two-step training, both the overall accuracy and the accuracy on the minority functions are generally better than using standard training and naïve two-step training.

- Latent Attention outperforms standard attention when using the same training method.

- The best Latent Attention model (Dict+LA) with two-step training can achieve $82.71\%$ and $64.84\%$ accuracy for trigger function on the gold test set, when trained on the SkewTop100 and SkewNonTop100 datasets respectively. For comparison, when using the entire training dataset, trigger function accuracy of Dict+LA is $89.38\%$. Note that the SkewNonTop100 dataset accounts for only $15.73\%$ of the entire training dataset.

- For SkewTop100 training set, Dict+LA model can achieve $78.57\%$ accuracy on minority functions in gold test set. This number for using the full training dataset is $85.12\%$, although the non-top100 recipes in SkewTop100 make up only $30.54\%$ of those in the full training set.

Correct Predictions

**(a)**

| | Post | your | Instagram | photos | to | Tumblr |
|---|---|---|---|---|---|---|
| latent | | | | | 0.75 | 0.14 |
| trigger | | | 0.8 | | | |
| action | | | | | | 0.76 |

| label | |
|---|---|
| trigger | Instagram.Any_new_photo_by_you |
| action | Tumblr.Create_a_photo_post |

**(b)**

| | Spreadsheet | with | the | daily | weather | , | triggered | at | sunrise. |
|---|---|---|---|---|---|---|---|---|---|
| latent | | | | | | 0.57 | | | |
| trigger | | | | | 0.15 | | 0.21 | | 0.47 |
| action | 0.33 | | | | | | 0.54 | | |

| label | |
|---|---|
| trigger | Weather.Sunrise |
| action | Google_Drive.Add_row_to_spreadsheet |

**(c)**

| | Instagram | > | flickr |
|---|---|---|---|
| latent | 0.12 | 0.81 | |
| trigger | 0.67 | 0.2 | 0.1 |
| action | 0.16 | 0.13 | 0.7 |

| label | |
|---|---|
| trigger | Instagram.Any_new_photo_by_you |
| action | Flickr.Upload_public_photo_from_URL |

**(d)**

| | If | send | IFTTT | a | text | tagged | #todo, | from | cell | phone | then | quick | add | event | to | google | calendar. |
|---|---|---|---|---|---|---|---|---|---|---|---|---|---|---|---|---|---|
| latent | | | | | | | | 0.16 | | | | 0.42 | | | 0.17 | | |
| trigger | | | | | 0.15 | 0.29 | 0.23 | | | | 0.12 | | | | | | |
| action | | | | | | | | 0.1 | | | | 0.18 | 0.14 | | | | 0.23 |

| label | |
|---|---|
| trigger | SMS.Send_IFTTT_an_SMS_tagged |
| action | Google_Calendar.Quick_add_event |

Misclassified Examples

**(e)**

| | Download | any | photos | of | me | to | dropbox | |
|---|---|---|---|---|---|---|---|---|
| latent | 0.11 | | | | | 0.83 | | Truth (Trigger) |
| | | | | | | | | Facebook.You_are_tagged_in_a_photo |
| trigger | 0.44 | 0.19 | 0.24 | | | | | Prediction |
| action | 0.34 | | 0.18 | | | | 0.39 | Android_Photos.Any_new_photo |

**(f)**

| | Instagram | to | Wordpress | |
|---|---|---|---|---|
| latent | | 0.92 | | Truth (Action) |
| | | | | WordPress.Create_a_post |
| trigger | 0.85 | | | Prediction |
| action | | | 0.8 | WordPress.Create_a_photo_post |

Figure 5: Examples of attention weights output by Dict+LA. `latent`, `trigger`, and `action` indicate the latent weights and active weights for the trigger and the action respectively. Low values less than 0.1 are omitted.

# 7 Empirical Analysis of Latent Attention

We show some correctly classified and misclassified examples in Figure 5 along with their attention weights. The weights are computed from a Dict+LA model. We choose Dict+LA instead of BDL-STM+LA, because the BDLSTM embedding of each token does not correspond to the token itself only — it will contain the information passing from previous and subsequent tokens in the sequence. Therefore, the attention of BDLSTM+LA is not as easy to interpret as Dict+LA.

The latent weights are those used to predict the action functions. In correctly classified examples, we observe that the latent weights are assigned to the prepositions that determine which parts of the sentence are associated with the trigger or the action. An interesting example is (b), where a high latent weight is assigned to ",". This indicates that LA considers "," as informative as other English words such as "to". We observe the similar phenomenon in Example (c), where token ">" has the highest latent weight.

In several misclassified examples, we observe that some attention weights may not be assigned correctly. In Example (e), although there is nowhere explicitly showing the trigger should be using a Facebook channel, the phrase "photo of me" hints that "me" should be tagged in the photo. Therefore, a human can infer that this should use a function from the Facebook channel, called "You_are_tagged_in_a_photo". The Dict+LA model does not learn this association from the training data. In this example, we expect that the model should assign high weights onto the phrase "of me", but this is not the case, i.e., the weights assigned to "of" and "me" are 0.01 and 0.007 respectively. This shows that the Dict+LA model does not correlate these two words with the You_are_tagged_in_a_photo function. BDLSTM+LA, on the other hand, can jointly consider the two tokens, and make the correct prediction.

Example (h) is another example where outside knowledge might help: Dict+LA predicts the trigger function to be `Create_a_post` since it does not learn that Instagram only consists of photos (and low weight was placed on "Instagram" when predicting the trigger anyway). Again, BDLSTM+LA can predict this case correctly.

**Acknowledgements.** We thank the anonymous reviewers for their valuable comments. This material is based upon work partially supported by the National Science Foundation under Grant No. TWC-1409915, and a DARPA grant FA8750-15-2-0104. Any opinions, findings, and conclusions or recommendations expressed in this material are those of the author(s) and do not necessarily reflect the views of the National Science Foundation and DARPA.

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
