[Supplementary Material · supplemental.pdf]



Figure 6: BDLSTM Embedding

# A  BDLSTM and attention model details

## A.1  BDLSTM embedding

Recurrent neural networks have become popular for natural language processing tasks due to their suitability for processing sequential data. Given inputs $\mathbf{x}_1$ to $\mathbf{x}_J \in \mathbb{R}^n$, a RNN computes

$$\mathbf{h}_t = \tanh(\mathbf{W}_{xh}\mathbf{x}_t + \mathbf{W}_{hh}\mathbf{h}_{t-1} + \mathbf{b}_h)$$

where $\mathbf{h}_0$ is a zero vector, $\mathbf{W}_{xh}$ and $\mathbf{W}_{hh}$ are trained parameter matrices respectively of size $m \times n$ and $n \times n$, and $\mathbf{b}_h \in \mathbf{R}^m$ is used as a bias.

Long Short-Term Memory (LSTM) is a RNN variant which is better suited for learning long-term dependencies. Although several versions of it have been described in the literature, we use the version in Zaremba et al. [21] and borrow their notation here:

$$\begin{pmatrix} i \\ f \\ o \\ g \end{pmatrix} = \begin{pmatrix} \sigma \\ \sigma \\ \sigma \\ \tanh \end{pmatrix} \mathbf{T}_{2n,4n} \begin{pmatrix} \mathbf{x}_t \\ \mathbf{h}_{t-1} \end{pmatrix}$$
$$\mathbf{c}_t = f \odot \mathbf{c}_{t-1} + i \odot g$$
$$\mathbf{h}_t = o \odot \tanh(c_t)$$

Here, $\sigma$ is the sigmoid function, and $\odot$ denotes the element-wise multiplication. The *memory cells* $\mathbf{c}_t$ are designed to store information for longer periods of time than the hidden state.

We construct the bi-directional model with a forward LSTM which receives the input sequence in the original order, and a backward LSTM which receives the input sequence in the reverse order. The BDLSTM embedding is the concatenation of the output of the two. This structure is illustrated in Figure 6.

## A.2  Standard attention model

The standard attention model differs with Latent Attention in the way that there is only one layer of active attention. In particular, we have

**The attention layer.**  We compute the attention $a$ over the $J$ tokens with the following:

$$a = \mathbf{softmax}(u^T \mathtt{Embed}_{\theta_1}(X)).$$

$a$ has $J$ dimensions and $u$ is a $d$-dimensional trainable vector.

Figure 7: $F_1$ score for arguments prediction

**Output representation.** We use a third set of parameters $\theta_3$ to embed $X$, and then the final output, a $d$-dimension vector, is the weighted-sum of these embeddings using the active weights.

$$o = \texttt{Embed}_{\theta_2}(X)a$$

**Prediction.** We compute probabilities over the output class labels by a matrix multiplication followed by softmax:

$$\hat{f} = \mathbf{softmax}(Wo)$$

# B    Predicting Arguments

We provide a frequency-based method for predicting the function arguments as a baseline, and show that this can outperform existing approaches dramatically when combined with our higher-performance function name prediction. In particular, for each description, we first predict the (trigger and action) functions $f_t, f_a$. For each function $f$, for each argument $a$, and for each possible argument value $v$, we compute the frequency that $f$'s argument $a$ takes the value $v$. We denote this frequency as $Pr(v|f,a)$. Our prediction is made by computing

$$\mathrm{argmax}_v Pr(v|f,a).$$

Note that the prediction is made entirely based on the predicted function $f$, without using any information from the description.

We found that for a given function, some arguments may not appear in all recipes using this function. In this case, we give the value a special token, $\langle MISSING \rangle$; this is distinct from the case where the argument exists but its value has zero length (i.e., "").

We use the same setup as in Section 5. The results are presented in Figure 7. [3] does not present their results for arguments prediction, so we do not include it in Figure 7. We can observe that the results are basically consistent with the results for channel and function accuracy.

# C    Data statistics and numerical results

In this section, we provide concrete data statistics and results. The statistics for IFTTT dataset that we evaluated is presented in Table 1. The numerical values corresponding to Figure 2, 3, and 7 are presented in Table 2. The statistics for the data used in one-shot learning are presented in Table 3. The numerical results corresponding to Figure 4a and 4b are presented in Table 4.

|  | Training | Test (Gold) |
|---|---|---|
| # of trigger channels | 112 | 59 |
| # of trigger functions | 443 | 136 |
| # of action channels | 87 | 41 |
| # of action functions | 161 | 56 |
| # of recipes | 68,083 | 584 |

Table 1: Statistics for IFTTT dataset

Channel Accuracy for Ensembled Models (Fig. 2)

| Ensemble | 1 | 2 | 3 | 4 | 5 | 6 | 7 | 8 | 9 | 10 |
|---|---|---|---|---|---|---|---|---|---|---|
| Dict | 71.9 | 72.8 | 73.5 | 74.1 | 74.7 | 80.1 | 80.5 | 79.6 | 80.5 | 81.3 |
| Dict+A | 82.4 | 83.0 | 83.6 | 83.2 | 83.2 | 83.2 | 83.7 | 83.9 | 83.6 | 83.7 |
| Dict+LA | 87.3 | 87.7 | 88.5 | 87.7 | 87.7 | 87.3 | 87.0 | 86.4 | 86.4 | 87.5 |
| BDLSTM | 84.8 | 89.2 | 90.1 | 90.4 | 90.6 | 90.8 | 90.4 | 90.9 | 91.4 | 91.6 |
| BDLSTM+A | 89.2 | 90.4 | 90.4 | 89.7 | 90.4 | 90.4 | 90.8 | 90.9 | 90.8 | 91.1 |
| BDLSTM+LA | 89.6 | 89.9 | 90.2 | 90.4 | 90.8 | 90.8 | 90.9 | 90.9 | 91.4 | 91.6 |
| Dong et al. [3] | 81.4 | | | | | | | | | |
| Beltagy et al. [7] | 89.7 | | | | | | | | | |
| Quirk et al. [16] | 89.1 | | | | | | | | | |

Function Accuracy for Ensembled Models (Fig. 3)

| Ensemble | 1 | 2 | 3 | 4 | 5 | 6 | 7 | 8 | 9 | 10 |
|---|---|---|---|---|---|---|---|---|---|---|
| Dict | 71.6 | 74.7 | 74.7 | 75.9 | 76.0 | 76.0 | 75.7 | 75.7 | 76.0 | 76.4 |
| Dict+A | 74.0 | 76.0 | 75.9 | 76.0 | 76.4 | 75.7 | 76.5 | 77.6 | 77.2 | 77.2 |
| Dict+LA | 79.6 | 78.4 | 78.0 | 78.9 | 78.0 | 79.9 | 79.9 | 79.9 | 81.3 | 82.2 |
| BDLSTM | 78.6 | 81.8 | 81.5 | 82.4 | 84.1 | 85.4 | 85.6 | 86.0 | 85.8 | 85.4 |
| BDLSTM+A | 80.3 | 83.6 | 84.6 | 84.4 | 84.6 | 84.4 | 84.4 | 84.6 | 85.1 | 84.8 |
| BDLSTM+LA | 82.4 | 83.7 | 85.3 | 86.0 | 85.8 | 85.6 | 86.0 | 86.8 | 87.5 | 87.3 |
| Dong et al. [3] | 78.4 | | | | | | | | | |
| Beltagy et al. [7] | 82.5 | | | | | | | | | |
| Quirk et al. [16] | 71.0 | | | | | | | | | |

F1 Score for Arguments for Ensembled Models (Fig. 7)

| Ensemble | 1 | 2 | 3 | 4 | 5 | 6 | 7 | 8 | 9 | 10 |
|---|---|---|---|---|---|---|---|---|---|---|
| Dict | 70.9 | 72.6 | 72.4 | 72.6 | 72.7 | 72.7 | 72.6 | 72.4 | 72.9 | 72.9 |
| Dict+A | 72.6 | 73.2 | 73.1 | 73.2 | 73.2 | 73.0 | 73.4 | 73.4 | 73.4 | 73.5 |
| Dict+LA | 73.1 | 73.8 | 74.5 | 74.2 | 74.9 | 74.8 | 74.7 | 75.0 | 75.1 | 75.1 |
| BDLSTM | 73.2 | 75.0 | 75.8 | 76.0 | 76.0 | 76.1 | 76.5 | 76.4 | 76.4 | 76.4 |
| BDLSTM+A | 74.4 | 75.8 | 75.9 | 75.9 | 76.0 | 76.0 | 75.8 | 76.0 | 76.1 | 76.0 |
| BDLSTM+LA | 74.7 | 76.0 | 76.0 | 76.3 | 76.2 | 76.2 | 76.3 | 76.8 | 76.7 | 76.8 |
| Dong et al. [3] | 74.2 | | | | | | | | | |
| Quirk et al. [16] | 66.5 | | | | | | | | | |

Table 2: Numerical Results for Figure 2 3, and 7

|  | SkewTop100 | SkewNonTop100 |
|---|---|---|
| # of recipes | 61,341 | 10,707 |
| # of recipes in $S$ | 58,376 | 9,707 |
| # of recipes not in $S$ | 2,965 | 1,000 |

Table 3: Statistics for unbalanced training sets

|  | B+S | BA+S | DA+S | BL+S | DL+S | BA+2N | DA+2N | BL+2N | DL+2N | BA+2 | DA+2 | BL+2 | DL+2 |
|---|---|---|---|---|---|---|---|---|---|---|---|---|---|
| | | | | | SkewTop100 training set | | | | | | | | |
| All | 77.91 | 79.5 | 78.4 | 81.51 | 80.3 | 80.82 | 81.85 | 81.34 | 80.99 | 80.99 | 81.9 | 81.3 | 82.7 |
| NonTop100 | 57.74 | 60.1 | 67.3 | 60.71 | 69.1 | 74.4 | 76.79 | 75.6 | 76.19 | 75 | 77.4 | 76.2 | 78.6 |
| | | | | | SkewNonTop 100 training set | | | | | | | | |
| All | 47.09 | 51 | 52.9 | 52.91 | 56 | 59.59 | 63.87 | 61.13 | 63.87 | 60.62 | 62.8 | 62.7 | 64.8 |
| Top100 | 31.01 | 37 | 39.7 | 40.87 | 42.8 | 55.29 | 56.01 | 53.61 | 56.01 | 54.09 | 53.9 | 57.7 | 57.5 |

Table 4: Numerical Results For Figure 4a and 4b