[Reviews · NeurIPS 2016]

Reviewer 1

Summary

This paper presents an end-to-end approach to translate natural language into simple If-This-Then-That (IFTTT) programs. The two techniques used in this paper are bidirectional LSTM (which is not novel) and hierarchical attention model (which is novel). Both lead to significant improvements in accuracy over previous work.

Qualitative Assessment

On balance I quite like this paper, as it addresses an interesting problem, and applies two approaches (one of which is novel), both showing very large improvement over the only previous work addressing this problem. The one-shot learning scenario where the authors' novel model ("Hierarchical Attention") was best seems related enough to a common problem in machine learning (bootstrapping to new domains with limited training data) to be convincing. One area where the paper could be improved is the description of the Hierarchical Attention model. In particular, it should be made more clear that parameter matrix V contains weights which depend on absolute word position (if I understand it correctly). This could obviously be a problem if the text descriptions under consideration were not of fairly uniform length, which drastically limits the general applicability of the approach under its current formulation. It is also unclear why the authors did not compare against a model which combines attention with an LSTM encoding, a more typical usage of attention model which seems likely to be very effective for this task. This seems like an obvious way of combining the advantages of both of the approaches considered. Though there are areas which could definitely be improved, I am somewhat inclined to accept this paper as it (1) addresses an interesting and little-studied problem, (2) shows that direct neural net approaches can achieve much better results than the one previous work addressing this problem, and (3) introduces a new idea which, with some refinement, could be generally useful in limited-training-data situations.

Confidence in this Review

2-Confident (read it all; understood it all reasonably well)


Reviewer 2

Summary

his paper presents an LSTM approach to parsing natural language “if then else statements" into IFTTT recipes. The authors also introduce a hierarchical attention mechanism. They compare LSTM and hierarchical attention mechanism on Quirk’s dataset, outperforming Quirk’s results substantially with both approaches, and doing slightly better with the LSTM approach. They also show one shot (or 10 shot) learning experiments where some trigger channels appear only very rarely in the training set. Here hierarchical attention performs better.

Qualitative Assessment

I think the task the task of mapping textual IFTTT recipe descriptions to actual IFTTT recipes is very interesting, and I am actually surprised this hasn’t been done. Well, it has been done [1], but it’s only coming out in ACL 2016 this year. The hierarchical attention model is an interesting contribution, and the one-shot learning setup is good idea to measure the models ability to generalise. I think there should be more papers that investigate this setting. The paper is relatively well written. There are a two core shortcomings: - For the most part it feels like a "apply LSTM to task X” paper. The hierarchical attention model is interesting, but unless I see this working on several problems it’s hard to accept this paper as a “new model” contribution. - I found it was quite difficult to follow the hierarchical attention model exposition. I think the core problem is that a lot of the intuition in 4.2.1 isn’t really connected to the formalisation in 4.2.2. In 4.2.2 a lot of intermediate vectors are introduced without ever linking them to the notions introduced in 4.2.1. Maybe a network visualisation would help? Moreover, there are some inconsistencies in notation and indexing that led me astray (see minor comments below). That’s unfortunate because I’d assume that this model is the core contribution of the paper. [1] https://arxiv.org/abs/1601.01280v1

Confidence in this Review

2-Confident (read it all; understood it all reasonably well)


Reviewer 3

Summary

Authors tries to translate IFTTT recipes written in natural language to for programmable classes (trigger/action channel/function). Two trainable networks are tested. The first is usual bi-directional RNN(BDRNN). The second is hierarchical attention (HA) model, which uses 3-layer feed-forward style attention mechanism to determine where to attend. Single HA shows better accuracy than a single BDRNN, but ensemble of models show BDRNNs perform better than HAs. Authors also tested whether the model can classify classes with much less training samples. The test environment is similar to one-shot learning. Two-step approach, which uses different data to train different parts. Two-step HA shows reasonable performance on samples with few training examples.

Qualitative Assessment

First, translation of the IFTTT dataset will be very useful and can be applied to many similar applications. However, as mentioned in the paper, there are changes and missing samples of the dataset (Also, the data is still changing) implies this dataset may not be a good target. The authors need to try another dataset as [2] uses. Second, more discussions and experiments on sharing embedding matrices will help investigating the nature of HA. The reviewer recommends the comparison of two cases: sharing embedding matrices and not. Third, SkewTop20 and SkewNonTop20 habe a different number of training samples in S (major set). Some figures about HA and two-level HA will help clarify the idea.

Confidence in this Review

2-Confident (read it all; understood it all reasonably well)


Reviewer 4

Summary

In this paper, the authors applied deep learning models for automatic translation of natural language descriptions into corresponding IFTTT recipes. They investigated bi-directional LSTM and attention models. They used one-shot learning for new channels and functions which have a very few recipes.

Qualitative Assessment

The paper is not organized well, especially the last sections. There is no conclusion or discussion section. It doesn’t provide any required explanation to emphasize the contribution. Presentation of the paper is bad; model architecture and formula descriptions are not clearly presented. The paper was not written clearly, especially from the sections 4 to 7. There are grammatical mistakes, and it is very hard to understand. The explanation of figures and tables are not correctly represented, and the data provided in the tables is not accurately represented in the respective sections. Architecture diagram of the model is not provided for easy understanding. Symbols and notions are not used consistently. Symbols and notions are mixed up. Same symbols are used for different variables/parameters. Authors used non-standard terms (E.g., Depending layer, Dependent layer, etc.). It is difficult to evaluate the originality of this work. The authors claim that there is only one existing work [14] on this problem. Novelty in the proposed models is not clearly explained. Formulization for HA is not clear in section 4.2.2. Moreover, it has only a small practical impact since this problem is more specific. The experimental settings differ for each model; hence comparisons among them are not valid. The data used for the base line model [14] and the proposed models are different. The authors have mentioned that many recipes used in [14] are no longer downloaded from IFTTT.com (section 5.1). Moreover, the experimental settings differ for the models HA and BDLSTM. Model evaluation methods are not described in detail. LSTM description differs from [19]. All the symbols and notations used are not explained in section 4.1. The formulations are not clear. Justifications are not provided for choosing BDLSTM. 1) In the abstract, it is mentioned as HA outperforms by 10 % but in the introduction it is mentioned as 11 points. (line no. 8 and 55) 2) Section - Introduction: Please justify why BDLSTM is chosen for comparison than the other models. 3) Please justify why 10 is chosen as number of models for average. 4) In Line no. 38, it is mentioned as BDLSTM improves over [14] by 15%. But Table 2. shows BDLSTM (average 10 models) improves over [14]. Please also check the same in the line nos. 48 to 51. 5) Table 2. shows HA (average 10 models) improves over [14] but in line no. 52 to 55, it is mentioned just HA. 6) The purpose of one-shot learning is not mentioned. Which setting it handles well? (line no. 56-58) 7) Section 4 - Model architecture: how variable-length descriptions are converted into a fixed-size vectors? Please explain. 8) Please check the equation of ht. The dimension of Whh ht-1 is not same as the other terms Wxh xt and bh. 9) The dimension of ht is not mentioned. 10) All the notations (symbols) used in the equations are not described. For e.g., i, f, o, g, etc. 11) Please use different symbol for i in wi (line no. 139) since i is used already. 12) Please check the equation of ct : dot product is wrong. It should be element-wise multiplication. 13) The symbols t and T are mixed up in the equations and throughout the section 4. Please use different symbols for Transpose, Affine transform and upper bound. 14) Please change pointwise production to element-wise multiplication (line no. : 131) 15) In line no. 136, order of g index is not correct. 16) Please explain the proposed model with the architecture diagram. How the models are built is not explained in detail. 17) How data is embedded to vector is not described? 18) In line no. 153, please cite and specify other works. 19) How importance of each word is measured and how weights are computed? (line no. 154). Please explain in detail. 20) The proposed Hierarchical Attention model is not explained clearly. Please explain the novelty of the proposed model ? 21) What are all the other “extra subtleties” does HA capture and how? (line no. 165) The authors provided only one example on “to”. 22) Please cite and specify “the usual attention mechanism”. (line no. 166) 23) Symbols are not consistently used for tokens: wi (line no. 139) and wT (line no. 173). 24) Depending layer, Dependent layer and Output representation are not standard terminologies. 25) The formulation in the sub-sub-section 4.2.2 is not clear. 26) Section 5 : In line no. 186, dependent -> hierarchical 27) Development set and Gold Test Set are not standard terminologies. 28) Table 2: Comparison of the model [14] with HA, HA (average 10 models), BDLSTM, BDLSTM (average 10 models) is not valid since the same dataset used by [14] is not used for this work. The authors say they are not able to download many recipes (line no. 191) as used in [14]. 29) The non-frequent tokens are mapped using < UNK >. How typos are handled; which are also not in the top 4000 most frequent tokens? (line no. 202) 30) Why the scaling factor is different for HA and BDLSTM? (line nos. 208 and 218) 31) Why the embedding vector sizes are different? (line nos. 215 and 221) 32) It is not clearly explained how over fitting is avoided. (line no. 209) 33) Error in representing the data presented in Table 2. (line nos. 230-231) 34) How will BDLSTM (average 10 models) be 10% better than any single trained model? 35) Line no. 236 : Performance of HA is not very much greater than BDLSTM 36) Evaluation methodologies are not described clearly. Which metrics are used for the evaluations of the models? 37) Which mistakes do HA and BDLSTM make? Please specify (line nos. 238-239) 38) It is mentioned the error rate is reduced by half. But it is not shown how. 39) Section 6 : Used same symbol for top-20 set and non-top-20 set 40) All the models mentioned in Table 4. are trained differently. Hence the comparison is not valid. 41) Please explain why it is difficult to have balance data for BDLSTM. 42) Table 4: Why for BDLSTM - Minor (NonTop20) for SkewTop20 is very low (4.01%)? 43) Section 7 is incomplete. 44) How HA matches the author’s expectations? (line no. 304) Minor comments: 1) Long sentence line no. 44-46 2) Please change points to percentage in lines 50, 55 3) Please change percentage points to percentage 4) Typo fo - of in line nos. 72 and 137 5) article ‘the’ is repeated : line nos. 168-169 6) English is not correct (line no. 175-178) – conjunctions are missed 7) Line no. 200 : such as characters - > characters such as 8) Present tense and past tense are mixed up. (e.g., line nos. 227, 228)

Confidence in this Review

2-Confident (read it all; understood it all reasonably well)


Reviewer 5

Summary

This paper proposed a hierarchical attention network for command recognition (the authors called it description-to-recipe translation task). The inputs are descriptions of commands, and the descriptions are encoded into vectors by the hierarchical attention network. The vector representations of commands are used to classify the commands into different categories (channels) and decide the types of actions need to be taken (functions). The bi-directional LSTM were also tested and compared to the proposed model. The authors reported that although the hierarchical attention networks generally performed worse than the bi-directional LSTMs, the former performed well under the situation that only a few samples are available for training while the latter may fail badly.

Qualitative Assessment

I had several troubles when reading the paper. My feeling is that the authors didn't represent their ideas in a clear way. (1) What did you mean by “we also observe that any single HA model outperforms the best BDLSTM model on any metric, but the results by averaging multiple models are not as good as BDLSTM”? To me, it is impossible. (2) Could you give more explanation of why the hierarchical attention networks performed better than the bi-directional LSTMs when only a few training samples are available? (3) Why did you train the model for 10 times and just report the best result (table 2)? I think the average of 10 results should be reported instead of the best one. (4) Whether s_j (in the equation below the line 180) and s_i (below the line 183) are referred to the same variable? It seems that they are vectors with different dimensionalities. (5) Where is the conclusion of the paper? (6) The notations, i, f, o, g, T, in the equations below the line 130 need to be explained.

Confidence in this Review

3-Expert (read the paper in detail, know the area, quite certain of my opinion)


Reviewer 6

Summary

This paper uses bi-directional LSTM model to translate natural language descriptions into If-This-Then-That (IFTTT) programs. It also designs a new model called hierarchical attention model, which is designed to directly capture the correct attention in a sentence to generate the recipe.

Qualitative Assessment

This paper use LSTM-based Neural Networks to learn the translation from natural language descriptions into If-This-Then-That (IFTTT) programs. This domain is interesting. It is very interesting and significant to let machine write codes according to human natural language. It is also a big challenge because natural language understanding and coding are both complex. However, the mapping patterns in IFTTT are not complicated. So it will be a good work if the authors use more complex programming language to test their work. It will be more explicit if authors use a figure to illustrate the architecture of Neural Networks used in the paper. The format of equations in this paper is not formal.

Confidence in this Review

2-Confident (read it all; understood it all reasonably well)